# Reliability and Validity of Self-Reported Questionnaires Assessing Physical Activity and Sedentary Behavior in Finland

**DOI:** 10.3390/ijerph21060686

**Published:** 2024-05-27

**Authors:** Pauliina Husu, Henri Vähä-Ypyä, Kari Tokola, Harri Sievänen, Paulo Rocha, Tommi Vasankari

**Affiliations:** 1The UKK Institute for Health Promotion Research, Kaupinpuistonkatu 1, FI-33500 Tampere, Finland; pauliina.husu@ukkinstituutti.fi (P.H.); henri.vaha-ypya@ukkinstituutti.fi (H.V.-Y.); kari.tokola@ukkinstituutti.fi (K.T.); harri.sievanen@ukkinstituutti.fi (H.S.); 2Portuguese Institute for Sport and Youth, 1250-190 Lisbon, Portugal; paulo.rocha@ipdj.pt; 3Faculty of Medicine and Health Technology, Military Medicine, Tampere University, Kalevantie 4, FI-33014 Tampere, Finland

**Keywords:** validity, reliability, accelerometry, self-report, test–retest study

## Abstract

Reliable and valid data on physical activity (PA) and sedentary behavior (SB) are needed for implementing evidence-based interventions and policies. Monitoring of these behaviors is based on PA questionnaires (PAQs) and device-based measurements, but their comparability is challenging. The present study aimed to investigate the test–retest reliability and concurrent validity of Finnish versions of the widely used PAQs (IPAQ-SF, EHIS-PAQ, GPAQ, Eurobarometer) and to compare their data with accelerometer data. This study is based on the Finnish data of the European Union Physical Activity and Sport Monitoring project (EUPASMOS). Participants (*n* = 62 adults, 62% women) answered the PAQs twice, one week apart, and wore an accelerometer for these seven consecutive days. Intraclass correlations, Spearman’s rank correlations, *t*-tests, and Cohen’s kappa with bootstrap confidence intervals were used to analyze the data. The PAQs had typically moderate-to-good test–retest reliability (ICC 0.22–0.78), GPAQ, EHIS-PAQ, and Eurobarometer showing the highest reliability. The PAQs correlated with each other when assessing sitting and vigorous PA (R = 0.70–0.97) and had a fair-to-substantial agreement when analyzing adherence to the PA recommendations (74–97%, Cohen’s kappa 0.25–0.73). All the PAQs had a poor criterion validity against the accelerometry data. The Finnish versions of the PAQs are moderately reliable and valid for assessing PA, adherence to PA recommendations and sitting among adult participants. However, the poor criterion validity against accelerometer data indicates that PAQs assess different aspects of PA constructs compared to accelerometry.

## 1. Introduction

A combination of low levels of physical activity (PA) and high sedentary behavior (SB) is recognized as a widely devastating global public health problem that requires immediate and effective action [1]. Physical inactivity, i.e., not meeting the current PA guidelines [2] is harmful to an individual’s health and wellbeing, increasing the risk for many adverse health conditions including coronary heart disease, type 2 diabetes, several cancers, anxiety, depression, cognitive impairments, and shortening of life expectancy [2]. SB is a distinct behavior from physical inactivity, defined as any waking behavior characterized by an energy expenditure of less than 1.5 metabolic equivalents (MET) while in a sitting or reclining posture [3]. SB is a risk factor for all-cause and cardiovascular mortality, incident cardiovascular disease, and type 2 diabetes [4], partly independently of PA [5,6].

The successful promotion of population health and wellbeing by increasing PA and decreasing SB requires reliable and valid prevalence data on these behaviors to enable the design, implementation, and evaluation of effective and cost-effective interventions and policies. Regular monitoring of PA and SB using reliable and valid methods on the population level is important for tracking the changes in these behaviors and for providing effective and reliable surveillance on which to base future policy decisions [2]. 

Traditionally, PA and SB have been assessed using self-reported methods like PA questionnaires (PAQs), but during the last decades, the use of device-based measurements has become popular in both intervention and population-based studies [7]. Both self-reported and device-based methods have strengths and limitations. Self-reported methods are cost-effective and feasible to be applied to large groups of individuals and have therefore been the basic method of assessing PA and SB in large epidemiological studies [8]. However, they are prone to recall bias, which typically leads to overestimation of PA [9] and underestimation of SB [10]. Respondents have especially faced challenges in reporting PA intensities, selecting different answer options, recalling instances of everyday activities, and calculating the total duration of more than one activity [11].

Device-based methods can identify the frequency, duration, and intensity (light, moderate, vigorous) of PA reliably, and they can also capture short intermittent bouts of activity that are impossible to measure using self-reported questionnaires [12,13]. Regarding SB, device-based methods can capture different bout lengths [14] and body postures [15]. However, device-based methods have a limited ability to distinguish between different domains of PA and SB (work, transportation, and recreational activities), a varying ability to detect specific types or modes of these behaviors, and are unable to identify contexts or individuals’ perceptions of them [12]. Furthermore, there are several device-based methods available with different device placements, wear protocols, and analysis methods and algorithms [12], which makes it challenging to compare studies with each other. Based on the above, no currently available method can assess all dimensions of PA and SB [8].

Despite the increasing use of device-based measurements, most EU countries still rely on PAQs when monitoring PA and SB nationally [8], and in most member states, these questionnaires have not been validated in their own language and cultural setting [16]. Since cultural context and country-specific types of PA may affect the interpretation of PAQs [17,18], and there are differences in the validity of the PAQs between different versions, all EU countries should validate the translated PAQs in their national settings [8]. The most widely used PAQs in population-based studies in the EU are the International Physical Activity Questionnaire-Short Form (IPAQ-SF), the Global Physical Activity Questionnaire (GPAQ), and the European Health Interview Survey-Physical Activity Questionnaire (EHIS-PAQ) [8]. According to a review and meta-analysis by Sember et al. [8], these questionnaires have been validated for moderate PA (MPA), vigorous PA (VPA), and moderate-to-vigorous PA (MVPA) in only 10 countries across the EU, not including Finland, where the current study was conducted. Furthermore, several studies have shown poor criterion validity of the PAQs against device-based measurements [19,20,21].

The purpose of the present study was to assess the 7-day test–retest reliability of Finnish versions of the IPAQ-SF, EHIS-PAQ, and GPAQ among adults; to compare various PAQ data to evaluate concurrent validity; and to compare the results of the PAQs with accelerometry to evaluate criterion validity. Additionally, the Eurobarometer questionnaire was included in the present study, since according to Sember et al. [8], there are no previous validation studies for the Eurobarometer PA questions.

## 2. Materials and Methods

The current investigation is part of the Finnish data of the European Union Physical Activity and Sport Monitoring project (EUPASMOS), which is an international study involving 18 European countries (EUPASMOS Plus 603328-EPP-1-2018-1-PT-SPO-SCP). In Finland, a convenience sample was drawn among 19–79-year-old adults nearby the city of Tampere, Finland (e.g., among the participants involved in another study of the UKK Institute and/or neighbors, relatives, or friends of the UKK Institute employees). No incentives were used. Altogether, 62 adults (62% women) participated in the study; answered the four IPAQ-SF, GPAQ, EHIS-PAQ, and Eurobarometer questionnaires twice, seven days apart and used an accelerometer (UKK RM42, UKK Terveyspalvelut Oy, Tampere, Finland) during these seven consecutive days. All the participants received written information about the purpose of the study, were instructed about the proper use of the accelerometer, and gave signed informed consent before participation. The study was carried out in accordance with the Declaration of Helsinki. The Regional Ethics Committee of the Expert Responsibility Area of Tampere University Hospital approved the study (R19031).

### 2.1. Questionnaires

#### 2.1.1. Questionnaires on PA

This study included Finnish versions of the IPAQ-SF, EHIS-PAQ, GPAQ, and Eurobarometer questions on PA and SB. IPAQ-SF, the first questionnaire developed for PA surveillance activities, has been implemented in several large surveillance programs both globally and in Europe [22] and is the most frequently used PAQ [19]. IPAQ-SF included seven items assessing the frequency and duration of walking, MPA, and VPA during the last 7 days.

GPAQ has been designed by the World Health Organization (WHO) for chronic disease risk factor surveillance and has been widely implemented [18,23]. It contains 16 items designed to assess the frequency and duration of PA in three domains (work, transportation, and leisure time) at two levels of intensity (MPA and VPA) during a typical week. It reports the PA duration by minutes per day and days per week for each PA domain and intensity level, which allows for the calculation of the energy expenditure and the metabolic equivalent of the task. Walking is not assessed by the GPAQ. 

EHIS-PAQ, created by Eurostat [24], provides data on the prevalence of PA and trends for cross-country comparisons [21]. EHIS-PAQ is used in the EU-wide surveillance system [21,25] and consists of eight items covering PA in three domains (work, transportation, and leisure time, including sport activities) and both aerobic and muscle-strengthening activities [25]. The activities are assessed across a typical week. In contrast to the other PAQs, the EHIS-PAQ does not cover the intensity of PA [25].

Eurobarometer is one of the tools used for decision-making in the EU [26] and is the most-used self-reported PAQ in European national surveillance systems [24]. The Eurobarometer assesses PA in terms of exercise or sports, other leisure-time activities, VPA, and MPA during the last 7 days, as well as walking. All the questions include several response options.

#### 2.1.2. Questionnaires on Sitting

All of the above-described PAQs include a question on sitting, indicating SB. The IPAQ-SF contains a question about sitting time during a workday during the last 7 days. The GPAQ asks about sitting and reclining time during waking hours on a typical day [27]. In both questionnaires, the participants indicate hours and minutes of sitting. The Eurobarometer includes a question about sitting, and the EHIS-PAQ includes a question about sitting and reclining on a typical day, with multiple choice response options. 

#### 2.1.3. Forward and Backward Method

The forward–backward method, slightly modified from Wild et al. [28], was used to translate the original PAQs from English to Finnish and back to English. In short, two independent translators translated the PAQs from the original English version into Finnish. The other two independent translators, one of which is native both in English and Finnish, translated them back into English. The two English versions were then compared with each other and with the original one. and the best translation was agreed upon. 

### 2.2. Accelerometer Measurement

Physical behavior in terms of time in bed, SB (lying, reclining, sitting), standing, light PA, MPA, and VPA was measured 24/7 using a tri-axial accelerometer (UKK RM42, UKK Terveyspalvelut Oy, Tampere, Finland). During waking hours, the accelerometer was worn on an elastic belt on the right side of the hip, excluding water-based activities. For the assessment of time in bed, the accelerometer was moved from the belt to an adjustable wristband and attached to the non-dominant wrist on the knuckles’ side. The accelerometer collected and stored the raw triaxial data in actual g-units in a ±16 G range at a 100 Hz sampling rate [29]. During waking hours, the mean amplitude deviation (MAD) was calculated based on the resultant acceleration signal in six-second epochs [30]. The epoch-wise MAD values were converted to METs, and PA intensity was calculated as the one-minute exponential moving average of epoch-wise MET values. Light PA was defined as MET values higher than or equal to 1.5 but less than 3.0 (MAD value between 22.5 and less than 91.5 mg), MPA was defined as MET values higher than or equal to 3.0 but less than 6.0 (MAD value at least 91.5 mg and less than 414 mg), and VPA was defined as MET values higher than or equal to 6.0 (MAD value at least 414 mg) [31]. The body posture (lying, reclining, sitting, standing) was determined for epochs during which the MAD values were lower than 22.5 mg [15].

The epoch-wise values representing lying, reclining, sitting, or standing periods were also smoothed by a one-minute exponential moving average. The determination of the body posture was based on the angle for posture estimation (APE) method, where the incident accelerometer orientation was assessed in relation to the reference gravity vector [15]. The posture was classified as standing if the APE was less than 11.6°, sitting if the APE was between 11.6° and 30.0°, reclining if the APE was between 30.0° and 73.0°, and lying otherwise [29].

To be included in the analysis, the participant had to have accelerometer data for at least 600 min per day during the waking hours. The participants were required to wear the accelerometer for at least 4 days during the seven-day measurement period, and any 4 days with at least 600 min of waking measurement time were applied [32]. Table 1 shows the variables assessed using each PAQ and the accelerometer. 

### 2.3. Statistical Analysis

The test–retest reliability of each questionnaire was assessed using intraclass correlation (ICC) of the single measures with 95% confidence intervals (CI). A two-way mixed effects model with an absolute agreement definition was used for the ICC calculation. An ICC less than 0.5 was considered poor, an ICC of 0.5–0.75 was moderate, an ICC 0.75–0.9 was good, and an ICC exceeding 0.9 was excellent [33]. The mean differences between the two measurements were analyzed using paired samples *t*-tests with a two-sided *p*-value. 

Concurrent validity was assessed in terms of the agreement between the four PAQs to assess different behaviors. Criterion validity was assessed as the agreement between the PAQs and the accelerometer-measured times of sitting, MPA, and VPA. Both validities were analyzed using Spearman’s rank correlations and the mean difference and 95% CI of the first measurement of the PAQs.

The agreement between the PAQs and the accelerometer to identify those participants who met the PA guidelines was assessed based on the percentage of agreement and Cohen’s kappa. Kappa values of 0.21–0.40 were interpreted as fair, those of 0.41–0.60 were moderate, 0.61–0.80 were substantial, and 0.81–1.00 were almost in perfect agreement [34]. Bootstrapping with 10,000 samples was used to estimate bias-corrected accelerated CIs for the *t*-tests, Spearman’s correlations and Cohen’s kappa. All the analyses were conducted using SPSS version 29 (IBM Corp. 2020, Armonk, NY, USA).

## 3. Results

The mean age of the participants was 47 years, and 62% of them were females (Table 2). The males were, on average, slightly older than the females. The mean body mass index (BMI) of the participants was 23.9 [standard deviation (SD) 2.5] kg/m^2^. Most of the participants were a healthy weight (BMI 18.5–24.9 kg/m^2^), and less than one third were overweight (BMI ≥ 25.9 kg/m^2^). Only one female had a BMI of less than 18.5 kg/m^2^, indicating that she was underweight. The participants wore the accelerometer for an average of 6.9 days during a week and an average of 15.3 (SD 2.2) hours per day during waking hours [males 15.3 (SD 1.2) and females 15.3 (SD 2.5) hours].

Table 3 describes the mean minutes (min) or hours and minutes (h:min) spent engaging in different PA intensities, walking, and sitting assessed using the four PAQs at the two measurement points one week apart and using an accelerometer. Regarding the PAQs, the mean minutes of MPA were the highest based on the GPAQ and the lowest based on the Eurobarometer at both measurement points. Also, the mean minutes of VPA were the highest with the GPAQ, but the differences between the PAQs were smaller for MPA. The sitting times were the highest with the EHIS-PAQ and the lowest with the IPAQ-SF at both measurement points. The IPAQ-SF provided the highest minutes for walking. Based on the accelerometer, the sitting time was higher and the minutes of VPA were especially lower than according to the PAQs. 

### 3.1. Test–Retest Reliability

The ICC values for each behavior are shown in Table 4. Within the 7-day interval, the ICC ranged from 0.22 (walking assessed using IPAQ-SF) to 0.78 (sitting assessed using EHIS-PAQ). Thus, the EHIS-PAQ provided good test–retest reliability (ICC > 0.75) for assessing sitting, and the GPAQ provided good test–retest reliability for assessing VPA and MVPA. All the other behaviors assessed using GPAQ, EHIS-PAQ and Eurobarometer provided moderate ICC values. The ICC values of the IPAQ-SF were poor (<0.5) for all the behaviors. The assessment of the mean differences between the two measurements had large CIs and high *p*-values, indicating no statistically significant differences between the test–retest measurements.

### 3.2. Concurrent and Criterion Validity

The assessment of sitting showed the highest correlations between the PAQs, with GPAQ vs. Eurobarometer and EHIS-PAQ vs. GPAQ having the highest correlations with each other (Table 5). Thus, reporting the sitting time using PAQs gives comparable results, regardless of the questionnaire. However, the accelerometer-measured sitting time differed from those of the PAQs. For sitting, the lowest correlation was found between the Eurobarometer and accelerometer. Regarding PA, both MPA and VPA showed higher correlations between the questionnaires compared to those observed between any PAQ vs. the accelerometer. VPA showed slightly higher correlations between the PAQs than MPA. The highest correlations for both MPA and VPA were found between the IPAQ-SF and Eurobarometer. When MPA and VPA were combined to MVPA, the GPAQ and Eurobarometer had the highest correlation with each other.

Table 6 presents the mean differences between the PA assessment tools in more detail. The accelerometer collected 40–90 min more of sitting per day than the PAQs, and the difference was statistically significant for all the PAQs except for the EHIS-PAQ. The IPAQ-SF and Eurobarometer did not differ from each other for assessing sitting and VPA. The EHIS and GPAQ did not differ from each other for sitting. The IPAQ-SF and GPAQ, on the other hand, did not differ from each other for assessing VPA. The accelerometer seemed to measure all physical behaviors differently compared to the PAQs. The only exception was the measurement of sitting assessed using the EHIS-PAQ and accelerometer, where the difference between the measurements did not reach statistical significance.

When assessing the adherence to PA guidelines using the four PAQs and the accelerometer, the EHIS-PAQ showed a 97% agreement with the GPAQ and an 86% agreement with the IPAQ-SF (Table 7). Also, the agreements of the GPAQ and EHIS-PAQ with the Eurobarometer were 95%, indicating a nearly perfect agreement. The lowest agreement percentage was found between the IPAQ-SF and accelerometer (74%). In line with these results, the EHIS-PAQ and GPAQ had a nearly perfect agreement, also according to the kappa values. The kappa values between the IPAQ-SF and accelerometer and between the Eurobarometer and accelerometer were negative, indicating no agreement at all.

## 4. Discussion

The present study evaluated the test–retest reliability and concurrent validity of widely used PAQs among Finnish adults. In addition, the criterion validity against accelerometry was assessed. The main results of this study indicate that the commonly used PAQs had typically moderate-to-good test–retest reliability over a one-week period. The reliability was the highest for the GPAQ, EHIS-PAQ, and Eurobarometer. The PAQs showed high correlations for both sitting time and VPA. Furthermore, the agreement between the PAQs was high when assessing their ability to measure the adherence to the physical activity recommendations. The highest agreements were found between the EHIS-PAQ and GPAQ, the EHIS-PAQ and Eurobarometer, and the GPAQ and Eurobarometer, indicating good concurrent validity. The PAQs collected 40–90 min less sitting and 27–81 min more MVPA per day than the accelerometer. Therefore, the accelerometer measurements of sitting and PA differed from those of the questionnaires, indicating a poor criterion validity for the PAQs, which is in line with previous studies [8,19,20,21]. Evidently, the PAQ and accelerometer measurements measure different aspects of physical behavior; PAQs are more precise in assessing the context and type of PA and SB, while accelerometers are more accurate in capturing the actual amount, intensities and bout lengths of PA and body postures of SB.

The present findings are in accordance with previous studies conducted in other EU countries. A Slovenian study found sitting and VPA to be the most valid and reliable constructs in all the tested PAQs, but the criterion validity of these constructs against the accelerometer was low [35], which was confirmed by the present study as well. A Hungarian study reported a weak-to-moderate criterion validity between the IPAQ-SF and accelerometer, but the test–retest reliability of the questionnaire was excellent [36]. Also, other studies have reported moderate-to-good test–retest reliability for the IPAQ-SF [17], GPAQ [20], and EHIS-PAQ [21]. However, in the present study, the test–retest reliability of the IPAQ-SF was only nearly moderate and slightly poorer than that of the other PAQs. 

According to Meh et al. [35], the GPAQ showed the highest criterion validity for VPA, and the EHIS-PAQ had the best agreement with the accelerometer data [37]. However, in the present study, the EHIS-PAQ showed a poor correlation with accelerometer-measured sitting, and we did not analyze the validity of MPA and VPA using the EHIS-PAQ, since it is not based on the perceived intensity of these activities but instead focuses on the description of the activities [11]. In line with the present findings, a meta-analysis by Sember et al. [8] assessing the test–retest reliability and concurrent and criterion validity of the IPAQ-SF, GPAQ, and EHIS-PAQ reported moderate-to-high reliability and moderate-to-high concurrent validity of the questionnaires against each other but only low-to-moderate criterion validity against device-based measurements.

In the present study, the participants reported less sitting using the PAQs when compared to the accelerometer-measured sitting times. Regarding MPA, the results varied between the PAQs. On average, the participants reported less MPA using the IPAQ-SF and Eurobarometer compared to the accelerometer, but the GPAQ yielded more MPA minutes than the accelerometer. Regarding both VPA and MVPA, the participants reported more activity minutes across all the PAQs than with the accelerometer. Studying only sitting, Dall et al. [38] reported a poor accuracy for 32 assessment tools, including the IPAQ and GPAQ, with under- and overestimation of the total sitting time and large limits of agreement. Also, Charles et al. [27] reported that the GPAQ underestimated sitting when compared to an accelerometer. A recent meta-analysis by Meh et al. [39] reported that the SB question of the IPAQ-SF and GPAQ had a low criterion validity and moderate-to-high test–retest reliability, which is similar to the reliability of the PA constructs in the same questionnaires [8]. According to a meta-analysis by Bakker et al. [40], single-item questionnaires can be recommended to assess self-reported SB because they show a similar validity and reliability compared to longer questionnaires. However, a more recent systematic review and meta-analysis by Meh et al. [39] recommend the use of multi-item SB questionnaires in PAQs together with device-based methods for collecting data on sitting rather than single-item SB questions. 

In line with the present findings, Meh et al. [35] reported that the participants underestimated sitting and overestimated MVPA across all the PAQs used compared to the accelerometer. The individuals with low levels of PA tended to especially overestimate their PA on the PAQs [41]. In studies by Meh et al., the GPAQ showed the highest criterion validity, especially for VPA [39], while the EHIS-PAQ agreed best with the accelerometer data in assessing MVPA [37]. Higher PA intensities are associated with more overestimation [42]. Meh et al. [37] found that overreporting seemed to be modified by physical fitness, since more-fit individuals were less prone to overreporting than less-fit ones. The less-fit participants perceived heavy breathing or increased heart rate at a lower PA intensity than the fitter individuals [37]. In addition to physical fitness, it has also been reported that the agreement between the methods is modulated by sex, age, weight [43], and education [44]. Thus, in future studies, these factors should be considered when analyzing and interpreting self-reported PA data. 

The differences between the PAQs may be partly due to differences in recall periods. While the EHIS-PAQ and GPAQ assess PA during a typical week, the IPAQ-SF and Eurobarometer assess the last 7 days. The discrepancy in the different time frames can impede the comparability, because PA during the last 7 days may not reflect the PA undertaken in a typical week [21]. Furthermore, the PAQs use different wording and definitions for behaviors and PA intensities, which may affect participants’ individual interpretations and responses. Major problems with PAQs have pertained to misunderstanding of PA intensities, classifying activities into different PA intensities, recalling instances of routine activities such as walking, cycling, and sitting, and combining the durations of activities performed in different domains and intensities for calculating the total amount of PA [11]. These problems are evident, especially among respondents aged 60 years or older [21].

The present study supports the findings of our previous study, where we concluded that the assessment of PA and SB is strongly method-dependent and not interchangeable [43]. Based on the present findings concurring with those of Cleland et al. [45], PAQs seem to function better on a group-level when assessing adherence to PA recommendations than on the individual level. For PAQs, the intensity of PA is assessed on a subjective scale, while with for accelerometers, the assessment of PA intensity is typically based on fixed thresholds [46].

The strengths of the present study include the use of four widely used PAQs, including the Eurobarometer survey, the validity of which has not been previously studied [8]. We used back-and-forth translations for the PAQs, which is a recommended way to conduct national validation studies. Furthermore, we used the recommended statistical parameters, kappa, and ICC values to analyze the reliability and validity of the PAQs [8]. Our sample included both sexes (62% female), which can also be seen as a strength of this study. A female majority has also been reported in previous studies [27,36,42]. The weaknesses of this study include a non-representative and relatively small convenience sample compared to other corresponding studies with approximately 100–300 participants [21,35,42,45]. A larger sample size would likely have given more accurate results, for example, in terms of narrower CIs. However, according to Altman [47], a sample size of at least 50 is considered adequate for a method comparison study. The other limitations of this study are its inability to take physical fitness or other background characteristics into account in the analysis and the potential impact of excluding water-based activities from the accelerometer data. 

## 5. Conclusions

The four commonly used PAQs evaluated in the present study showed typically moderate-to-good test–retest reliabilities. They correlated with each other when assessing the sitting time and VPA and had a high agreement when analyzing the adherence to the PA recommendations. All these findings indicate a good concurrent validity and the ability of the PAQs to assess PA and SB on a group level. However, the poor agreement with the accelerometer measurements denotes a poor criterion validity. The present findings confirm the notion that the PAQs and accelerometers assess different aspects of PA and cannot be used interchangeably.

## Figures and Tables

**Table 1 ijerph-21-00686-t001:** The variables assessed using the PAQs and the accelerometer.

	MPA	VPA	Walking	Sitting	MVPA *
IPAQ-SF	X	X	X	X	X
GPAQ	X	X	na	X	X
Eurobarometer	X	X	X	X	X
EHIS-PAQ	na	na	X	X	na
Accelerometer	X	X	na	X	X

na = not applicable, MPA = moderate physical activity, VPA = vigorous physical activity, MVPA = moderate-to-vigorous physical activity. * IPAQ-SF and Eurobarometer: MVPA = MPA + VPA + walking; GPAQ and accelerometer: MVPA = MPA + VPA.

**Table 2 ijerph-21-00686-t002:** Background characteristics of the participants.

	Men (n = 23)	Women (n = 38)	Total (n = 62) *
Age, years (SD)	48.9 (15.3)	46.7 (16.5)	47.1 (16.2)
Height, cm (SD)	179.6 (5.9)	166.3 (5.9)	171.2 (8.7)
Weight, kg (SD)	79.3 (8.5)	65.2 (7.4)	70.3 (10.4)
BMI < 18.5 kg/m^2^, %	0	2.6	1.6
BMI 18.5–24.9 kg/m^2^, %	65.2	71.1	69.4
BMI ≥ 25.0 kg/m^2^, %	34.8	26.3	29.0

* One participant did not report sex. BMI = body mass index, SD = standard deviation.

**Table 3 ijerph-21-00686-t003:** Mean minutes (min) or hours and minutes (h:min) with standard deviation (SD) spent engaging in different behaviors according to the four PAQs at two measurement points and according to an accelerometer (n = 62).

		MPA	VPA	Walking	Sitting	MVPA *
		min (SD)	min (SD)	min (SD)	h:min (SD)	h:min (SD)
IPAQ-SF	Measurement 1	29 (29)	23 (22)	70 (129)	6:42 (2:35)	2:02 (2:16)
	missing (n)	-	-	-	1	-
	Measurement 2	28 (38)	21 (29)	47 (65)	6:39 (2:34)	1:36 (1:44)
	missing (n)	-	-	-	-	-
GPAQ	Measurement 1	67 (57)	28 (43)	na	7:19 (3:05)	1:35 (1:24)
	missing (n)	-	-	-	-	-
	Measurement 2	69 (69)	29 (52)	na	7:05 (2:42)	1:38 (1:49)
	missing (n)	-	-	-	-	-
Eurobarometer	Measurement 1	22 (22)	20 (21)	28 (21)	6:48 (2:10)	1:08 (0:39)
	missing (n)	1	1	1	1	-
	Measurement 2	23 (27)	20 (24)	29 (28)	6:55 (1:56)	1:10 (0:58)
	missing (n)	2	1	1	-	-
EHIS-PAQ	Measurement 1	na	na	21 (26)	7:29 (2:49)	na
	missing (n)			-	-	
	Measurement 2	na	na	21 (27)	7:52 (2:38)	na
	missing (n)			1	-	
Accelerometer	Measurement 1	41 (24)	4 (7)	na	8:12 (1:39)	0:41 (0:24)
	missing (n)	-	-		-	-

na = not applicable, MPA = moderate physical activity, VPA = vigorous physical activity, MVPA = moderate-to-vigorous physical activity. * IPAQ-SF and Eurobarometer: MVPA = MPA + VPA + walking; GPAQ and accelerometer: MVPA = MPA + VPA.

**Table 4 ijerph-21-00686-t004:** Test–retest reliability, intraclass correlation (ICC) of the average measures with 95% confidence intervals (95% CI), and mean differences between the two measurements with 95% CI and *p*-value.

			Difference	2-Sided
		ICC (95% CI)	Mean (95% CI) *	*p*-Value **
IPAQ-SF	MPA	0.45 (0.23; 0.63)	1.4 (−7.8; 10.0)	0.771
	VPA	0.45 (0.23; 0.63)	1.5 (−5.7; 8.0)	0.683
	Walking	0.22 (−0.03; 0.44)	23.2 (−3.0; 61.3)	0.282
	Sitting	0.49 (0.27; 0.66)	3.0 (−34.4; 43.8)	0.883
	MVPA ***	0.36 (0.13; 0.56)	26.0 (−3.9; 64.6)	0.219
GPAQ	MPA	0.61 (0.43; 0.75)	−2.0 (−16.9; 10.8)	0.778
	VPA	0.77 (0.64 0.85)	−1.1 (−10.2; 5.9)	0.809
	Sitting	0.50 (0.29; 0.67)	14.4 (−25.8; 59.9)	0.536
	MVPA ***	0.77 (0.64; 0.85)	−3.1 (−20.5; 12.2)	0.719
Eurobarometer	MPA	0.52 (0.31; 0.69)	−1.5 (−8.0; 4.9)	0.659
	VPA	0.72 (0.57; 0.82)	1.0 (−3.5; 5.4)	0.667
	Walking	0.56 (0.35; 0.71)	−0.6 (−6.9; 5.4)	0.850
	Sitting	0.74 (0.60; 0.83)	−8.6 (−33.2; 16.1)	0.496
	MVPA ***	0.65 (0.48; 0.77)	−1.1 (−12.7; 9.8)	0.847
EHIS-PAQ	Walking	0.66 (0.49; 0.78)	−0.2 (−5.7; 5.2)	0.959
	Sitting	0.78 (0.65; 0.86)	−23.6 (−51.1; 2.0)	0.092

MPA = moderate physical activity, VPA = vigorous physical activity, MVPA = moderate-to-vigorous physical activity. * paired samples *t*-test with 95% bootstrap confidenceinterval. ** Bootstrap 2-sided *p*-value. *** IPAQ-SF and Eurobarometer: MVPA = MPA + VPA + walking; GPAQ and accelerometer: MVPA = MPA + VPA.

**Table 5 ijerph-21-00686-t005:** Spearman’s rank correlations between the four PAQs vs. the accelerometer.

	Sitting	MPA	VPA	Walking	MVPA ˚
IPAQ-SF–Eurobarometer	0.873 **	0.599 **	0.783 **	0.458 **	0.611 **
IPAQ-SF–GPAQ	0.875 **	0.484 **	0.723 **	na	0.622 **
IPAQ-SF–EHIS-PAQ	0.878 **	na	na	0.307 *	na
IPAQ-SF–Accelerometer	0.324 *	0.048	0.204	na	0.217
GPAQ–Eurobarometer	0.965 **	0.357 **	0.704 **	na	0.699 **
GPAQ–Accelerometer	0.256	0.337 **	0.309 **	na	0.220
Eurobarometer–Accelerometer	0.193	−0.061	0.325 **	na	0.212
EHIS-PAQ–Eurobarometer	0.928 **	na	na	0.569 **	na
EHIS-PAQ–Accelerometer	0.257	na	na	na	na
EHIS–PAQ-GPAQ	0.964 **	na	na	na	na

na = not applicable, MPA = moderate physical activity, VPA = vigorous physical activity, MVPA = moderate-to-vigorous physical activity. ˚ IPAQ-SF and Eurobarometer: MVPA = MPA + VPA + walking; GPAQ and accelerometer: MVPA = MPA + VPA. Significance of bootstrap CIs * <0.05, ** <0.01.

**Table 6 ijerph-21-00686-t006:** Mean difference and 95% bootstrap confidence intervals (95% CI) between the four PAQs and the accelerometer. Bolded values indicate non-statistically significant differences between the measurement tools.

		Sitting	MPA	VPA	Walking	MVPA *
IPAQ-SF–Eurobarometer	mean	**−6.5**	8.5	**2.6**	43.9	54.4
	95% CI	**−24.7; 11.8**	3.1; 14.2	**−0.9; 6.1**	17.4; 79.8	27.8; 91.3
IPAQ-SF–GPAQ	mean	−34.4	−38.4	**−5.4**	na	27.1
	95% CI	−63.9; −9.3	−51.6; −26.4	**−13.9; 1.6**		0.5; 64.4
IPAQ-SF–EHIS-PAQ	mean	−49.5	na	na	50.4	na
	95% CI	−71.1; −29.7			24.0; 86.3	
IPAQ-SF–Accelerometer	mean	89.6	11.5	−18.7	na	−81.0
	95% CI	51.5; 130.8	2.9; 20.2	−24.3; −13.4		−120.0; −51.5
GPAQ–Eurobarometer	mean	28.0	46.9	8.0	na	27.4
	95% CI	8.5; 54.1	34.2; 60.9	0.7; 17.1		13.0; 43.6
GPAQ–Accelerometer	mean	55.2	−26.9	−24.1	na	−54.0
	95% CI	2.8; 105.2	−41.2; −13.8	−35.9; −14.6		−75.5; −35.0
Eurobarometer–Accelerometer	mean	83.2	20.0	−16.1	na	−26.6
	95% CI	44.6; 123.3	12.1;27.9	−21.4; −11.3		−37.2; −16.4
EHIS-PAQ–Eurobarometer	mean	−43.0	na	na	6.5	na
	95% CI	−62.4; −23.6			1.9; 10.9	
EHIS-PAQ–Accelerometer	mean	**40.1**	na	na	na	na
	95% CI	**−6.9; 87.3**				
EHIS-PAQ–GPAQ	mean	**−15.1**	na	na	na	na
	95% CI	**−28.9; 1.5**				

na = not applicable, MPA = moderate physical activity, VPA = vigorous physical activity, MVPA = moderate-to-vigorous physical activity. * IPAQ-SF and Eurobarometer: MVPA = MPA + VPA + walking; GPAQ and accelerometer: MVPA = MPA + VPA.

**Table 7 ijerph-21-00686-t007:** Meeting the physical activity recommendations according to the four PA questionnaires and the accelerometer.

	Agreement (%)	Cohen’s kappa	*p*-Value	95% CI *
IPAQ-SF–Eurobarometer	87.1	0.371	0.001	0.000; 0.664
IPAQ-SF–GPAQ	88.7	0.477	<0.001	0.136; 0.763
IPAQ-SF-EHIS–PAQ	85.5	0.252	0.015	−0.031; 0.546
IPAQ-SF–Accelerometer	73.8	−0.041	0.750	−0.194; 0.193
GPAQ–Eurobarometer	95.2	0.641	<0.001	0.000; 1.000
GPAQ–Accelerometer	82.0	0.059	0.634	−0.127; 0.326
Eurobarometer–Accelerometer	80.3	−0.096	0.421	−0.175; 0.000
EHIS-PAQ–Eurobarometer	95.2	0.546	<0.001	−0.025; 1.000
EHIS-PAQ–Accelerometer	85.2	0.119	0.287	−0.085; 0.408
EHIS-PAQ–GPAQ	96.8	0.734	<0.001	0.000; 1.000

* 95% bootstrap confidence interval.

## Data Availability

The data are maintained at the UKK Institute. The datasets analyzed in the present study are not publicly available due to ethical restrictions (the Regional Ethics Committee of the Expert Responsibility Area of Tampere University Hospital), but more detailed information on the data is available from the corresponding author upon reasonable request.

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
