# Peer review of "Reliability and Validity of Self-Reported Questionnaires Assessing Physical Activity and Sedentary Behavior in Finland"

_ijerph, 2024, doi:10.3390/ijerph21060686_

Round 1

Reviewer 1 Report

Comments and Suggestions for Authors

The authors aimed to investigate test-retest reliability and concurrent validity of Finnish versions of IPAQ-SF, EHIS-PAQ, GPAQ, and Eurobarometer and compare their data with accelerometer data. This manuscript was interesting to review, but the methods require further clarification and the discussion requires expansion. Detailed comments are below:

INTRODUCTION:

-        Lines 61-64: Please provide references for your statements that “…device-based methods have limited ability to distinguish between different domains of PA and SB (work, transportation, and recreational activities), varying ability to detect specific types or modes of these behaviors and are unable to identify contexts or individual’s perception of them.”

-        Please outline the reasons why it is challenging to compare studies using different device brands.

-        It was unclear why your introduction discussed the differing PA devices domains and devices and how this relates to your study.

-        It is unclear as to why the translation impacts on the PA and SB measures and why your study is important here.

MATERIALS AND METHODS:

-        Please clarify the source(s) of your convenience sample and nature of incentive

-        Please outline your power analysis to justify your sample size

-        Lines 107-108: Please correct to past tense. Also, please outline to the reader your criteria for “poor criterion validity”.

-        Lines 108-109: Please correct to represent this statement relates to prior work

-        Line 113: Define MPA and VPA to the reader

-        Lines 117-118: Please outline to the reader your criteria for “very good test–retest reliability” and “validity … has been poor or fair”

-        Lines 124-126: Please outline to the reader your criteria for “Test-retest reliability of EHIS-PAQ has been moderate to high, but validity against device-based measurement is low”

-        Line 137: Please outline to the reader your criteria for “… low criterion validity and moderate to high test–retest reliability”

-        Line 172: Please outline why 4 days out of the 7 was chosen

-        Line 178: Please check the English for this statement.

-        Statistical analysis: Please justify why you have not presented bootstrapped CI for your Kappa

RESULTS

-        Please outline the metrics and profile of participants related to how long they wore the accelerometer as it was mentioned in your methods that “Participants were required to wear the accelerometer for at least 4 days during the seven-day measurement period.” Also, what days were covered as this may confound your results.

-        Line 193: Please present SD when reporting means

-        Lines 196-197: Please articulate the proportion underweight, normal, obese classes I to III per sex

-        Table 1: It was unclear why GPAQ and EHIS-PAQ contained ‘na’ measurements as this was not outlined in the methods

-        Tables 2 and 3: The 95% CI values are very broad for a number of your measures. Please explain these very large CI’s

-        Tables 2 and 3: Please clarify in your methods why certain activities are missing from your metrics. It may be beneficial in your methods to include a simple table clarifying to the reader your measures and what is/is not covered by them.

-        Table 3: Please include your diagnostics to ensure that your data meets the assumptions for correlation and that there were no influencing observations.

-        Table 5: Please include bootstrapped CI’s for your Kappa

DISCUSSION:

-        Line 272: Please correct your English.

-        Please include your limitations such as you collected a convenience sample yet your statistical methods assumed a simple random sample was collected.

-        Please outline the potential impact of excluding water-based activities

-        It was unclear in your Discussion as to the impact of your results as this section was predominantly discussing the similarities/dissimilarities of your results with other studies.

Comments on the Quality of English Language

Please check your English as there were occasions you had used the incorrect tense or provided sentences that needed reviewing.

Author Response

Thank you for the valuable comments for our manuscript. The comments were carefully checked through and the following list will clarify the modifications made. In the revised manuscript file, the changes are marked with MS word track changes. We hope that these revisions improve the quality of our manuscript so that it becomes acceptable for publication in the IJERPH. However, if we have not covered your comments carefully enough, we are willing to modify the manuscript further.

Reviewer 1

Comments and Suggestions for Authors

The authors aimed to investigate test-retest reliability and concurrent validity of Finnish versions of IPAQ-SF, EHIS-PAQ, GPAQ, and Eurobarometer and compare their data with accelerometer data. This manuscript was interesting to review, but the methods require further clarification and the discussion requires expansion. Detailed comments are below:

INTRODUCTION:

-        Lines 61-64: Please provide references for your statements that “…device-based methods have limited ability to distinguish between different domains of PA and SB (work, transportation, and recreational activities), varying ability to detect specific types or modes of these behaviors and are unable to identify contexts or individual’s perception of them.”

  • Thank you for pointing this out. We have added references Sattler et al. (2021) and Dowd et al. (2018)

-        Please outline the reasons why it is challenging to compare studies using different device brands.

  • Thank you for asking clarification for this. We have replaced the word brand by the word method, since different device-based methods use different device placement, wear protocols, and analysis methods and algorithms, which makes it challenging to compare studies with each other. This has been indicated in lines 64-67.

-        It was unclear why your introduction discussed the differing PA devices domains and devices and how this relates to your study.

  • Thank you for noticing this. We have clarified this in lines 67-71. The different methods are discussed because they all have strengths and weaknesses, and no currently available method can assess all dimensions of PA and SB.

-        It is unclear as to why the translation impacts on the PA and SB measures and why your study is important here.

  • Thank you. This sentence has been re-written “Since cultural context and country-specific types of PA may affect the interpretation of PAQs (Craig et al. 2003, Bull et al. 2009) and the validity of the PAQs may differ between different language versions, all EU countries should validate the translated PAQs in their national settings (Sember et al. 2020).

MATERIALS AND METHODS:

-        Please clarify the source(s) of your convenience sample and nature of incentive

  • Thank you for asking clarification for this. The sources of the sample have now been specified. No incentives were used, and this point has been added to the text (lines 97-99).

-        Please outline your power analysis to justify your sample size

  • Thank you for asking this important issue. Unfortunately, we did not perform any formal power analysis. The original purpose was to analyze data from several countries (about 70-80 participants per country) together which would have provided a larger sample size, but coexisting COVID-19 undermined the data collections and timetables in most countries, and therefore we decided to focus on the Finnish data with a sample size of 62 adults. Sample sizes of 40 or higher providing a broad variation in data are generally considered adequate to reveal meaningful differences between two different methods.

-        Lines 107-108: Please correct to past tense. Also, please outline to the reader your criteria for “poor criterion validity”.

  • Thank you for noticing this. The tense has been corrected. According to the comments of the reviewer 2, these sentences have been modified and removed to discussion.

-        Lines 108-109: Please correct to represent this statement relates to prior work

  • Thank you for pointing out this inconsistency. According to the comments of the reviewer 2, this has been modified and removed to introduction.

-        Line 113: Define MPA and VPA to the reader

  • Thank you for asking this. These abbreviations have been defined in line 82 when mentioned for the first time.

-        Lines 117-118: Please outline to the reader your criteria for “very good test–retest reliability” and “validity … has been poor or fair”

  • Thank you for asking more details here. According to the suggestions of the reviewer 2, these issues have been modified and removed to introduction and discussion.

-        Lines 124-126: Please outline to the reader your criteria for “Test-retest reliability of EHIS-PAQ has been moderate to high, but validity against device-based measurement is low”

  • According to the comments of the reviewer 2, these issues have been modified and removed to introduction and discussion.

-        Line 137: Please outline to the reader your criteria for “… low criterion validity and moderate to high test–retest reliability”

  • Thank you. According to the comments of the reviewer 2, these issues have been modified and removed to introduction and discussion.

- Line 172: Please outline why 4 days out of the 7 was chosen

  • Thank you for asking justification for this. We have added the reference Donaldson et al. (2016) who reported that 4 days are representative enough for accelerometer measurements (lines 188-189). In addition, we have added to the results that the participants wore the accelerometer for on average 6.9 days during a week (lines 225-226).

-        Line 178: Please check the English for this statement.

  • Thank you! This has been re-written and divided into two sentences (lines 199-201).

-        Statistical analysis: Please justify why you have not presented bootstrapped CI for your Kappa

  • Thank you. We have now performed the analysis with Bootstrap CIs for Cohen’s Kappa, Spearman’s Rank Correlations and T-tests.

RESULTS

-        Please outline the metrics and profile of participants related to how long they wore the accelerometer as it was mentioned in your methods that “Participants were required to wear the accelerometer for at least 4 days during the seven-day measurement period.” Also, what days were covered as this may confound your results.

  • Thank you for asking more detailed information here. The wear time has been added (lines 225-227). As stated in the methods any 4 days were applicable for the data as suggested by Donaldson et al. (2016), and the participants wore the accelerometer for on average 6.9 days during a week.

-        Line 193: Please present SD when reporting means

  • Thank you. SDs have been added.

-        Lines 196-197: Please articulate the proportion underweight, normal, obese classes I to III per sex

  • These proportions have been added to the text (lines 222-225).

-        Table 1: It was unclear why GPAQ and EHIS-PAQ contained ‘na’ measurements as this was not outlined in the methods

  • Thank you for asking clarification. GPAQ has been described in lines 120-126, and it does not include a question on walking (line 126). That is why GPAQ contains ‘na’ regarding walking in the tables.
  • It has been stated in lines 134-135 that in contrast to the other PAQs, EHIS-PAQ does not cover the intensity of PA. That is why it has ‘na’ regarding PA intensity in the tables. We have also produced a new table 1 to indicate the variables analyzed in different methods.

-        Tables 2 and 3: The 95% CI values are very broad for a number of your measures. Please explain these very large CI’s

  • Thank you for noticing this. Large CIs are based on the answers for different questions. Respondents’ interpretation about their PA and SB varies between the questionnaires and time points.

-        Tables 2 and 3: Please clarify in your methods why certain activities are missing from your metrics. It may be beneficial in your methods to include a simple table clarifying to the reader your measures and what is/is not covered by them.

  • Thank you. Please see our answer to a previous comment. We have also added a new table 1 to the methods as suggested.

-        Table 3: Please include your diagnostics to ensure that your data meets the assumptions for correlation and that there were no influencing observations.

  • We have switched correlations to Spearman’s Rank Correlations and calculated Bootstrapped significance levels for them (table 4).

-        Table 5: Please include bootstrapped CI’s for your Kappa

  • Thank you! Bootstrapped CIs have now been added.

DISCUSSION:

-        Line 272: Please correct your English.

  • Thank you for noticing incorrect language. This sentence has been re-written.

-        Please include your limitations such as you collected a convenience sample yet your statistical methods assumed a simple random sample was collected.

  • Thank you for asking clarification for this. Convenience sample has been mentioned as one of the limitations of the study (line 382). We have now calculated the Bootstrapped CIs which does not include any assumptions for distribution.

-        Please outline the potential impact of excluding water-based activities

  • Thank you for detecting this. This has been added as one of the weaknesses of the study.

-        It was unclear in your Discussion as to the impact of your results as this section was predominantly discussing the similarities/dissimilarities of your results with other studies.

  • Thank you. We aimed at discussing the similarities/dissimilarities of our results with other studies in the discussion. In the revised version we have emphasized the impact of our results in conclusions (lines 391-392).

Comments on the Quality of English Language

Please check your English as there were occasions you had used the incorrect tense or provided sentences that needed reviewing.

  • The English language has been checked throughout the manuscript.

Reviewer 2 Report

Comments and Suggestions for Authors

The study is assessing various translated physical activity questionnaires and accelerometer measurements for validity in assessing physical activity and sedentary behaviour as well as testing the intercorrelation of these methods to do so. While the results are especially of importance to Finland itself, they do have meaningful insight in the conclusive recommended use.

(1) However, the methods section should be given some attention: Make different sections: 2.1.1 Questionnaires on PA and 2.1.2 SB in the self-report questionnaires (from line 131 onwards); 2.1.3 Forward-backward method, from line 139 onwards. Many statements onder the questionnaires should be briefly mentioned in introduction, but more in detail used for the discussion, but certainly not part of the methods section.

(2) The authors should also think about presenting some of the tabulated data in figure format, for the reader so get a better insight of what specifically is being compared. That would certainly also facilitate the writing of the results section.

(3) In the discussion, the authors should address more in detail the low number of participants in relation to drawing general conclusion. As well as compare the number of participants to other comparable studies, and discuss the gender fraction and how that contributed to attitudes and interpretations.

More specific edits/suggestions are presented below:

Introduction:

Lines 30-33 should read: ‘Low levels of physical activity (PA) and high sedentary behavior (SB) are recognized combined as widely devastating global public health problems that require drastic actions [1]. Physical inactivity i.e., not meeting the current PA guidelines [2], is harmful to an individual’s health and wellbeing, and increases the risk for many adverse health conditions…’

Lines 37-39 should include a comma: ‘SB is a risk factor for all-cause and cardiovascular mortality, incident cardiovascular disease, and type 2 diabetes [4], partly independently of PA [5, 6].’

Add citations to lines 57/59.

Add citations to lines 61/64 and first part of 64/65 ‘…methods and algorhythms’.

Add citations to lines 68/69 ‘…when monitoring PA and SB nationally’

Add citation to lines 72/76.

Replace in line 79 ‘this’ for ‘the current’

Replace in line 81/82 ‘their data with each other’ by ‘various PAQ data’.

Write in line 83 ‘Additionally, the Eurobarometer questionnaire was included in the present study, since …’

Methods:

Line 87, replace: ‘The current investigation is part of the Finnish….’

Lines 106/107 ‘but IPAQ-SF seems to overreport PA when compared to device-based measurement, which indicates poor criterion validity [16]’, perhaps belongs in the discussion.

Lines 107/108 ‘VPA and walking showed the highest correlation with device-based measurements [16]’ should be part of the introduction.

Line 116 ‘Additionally, one item assesses sitting during a typical day [20].’ Should be under section 2.1.2.

Line 117 ‘GPAQ has shown moderate to very good test–retest reliability’ should be part of the discussion.

Lines 117/118 ‘but validity against device-based measurements has been poor or fair [21]’ should be part of the introduction.

Line 124 ‘Test-retest 124 reliability of EHIS-PAQ has been moderate to high’ should be part of the discussion.

Lines 125/126 ‘but validity against device-based measurement is low [23]’ should be part of the introduction.

Make different sections: 2.1.1 Questionnaires on PA and 2.1.2 SB in the self-report questionnaires (from line 131 onwards); 2.1.3 Forward-backward method, from line 139 onwards. Many statements onder the questionnaires should be briefly mentioned in introduction, but more in detail used for the discussion, but certainly not part of the methods section.

Results

Line 242 instead of ‘than the’ use ‘as compared to the’

Discussion

Line 261, remove ‘with each other’ to ‘PAQs showed good correlations for both sitting time and VPA’.

Author Response

Thank you for your valuable comments for our manuscript. The comments were carefully checked through and the following list will clarify the modifications made. In the revised manuscript file, the changes are marked with MS word track changes. We hope that these revisions improve the quality of our manuscript so that it becomes acceptable for publication in the IJERPH. However, if we have not covered your comments carefully enough, we are willing to modify the manuscript further.

Reviewer 2

Comments and Suggestions for Authors

The study is assessing various translated physical activity questionnaires and accelerometer measurements for validity in assessing physical activity and sedentary behaviour as well as testing the intercorrelation of these methods to do so. While the results are especially of importance to Finland itself, they do have meaningful insight in the conclusive recommended use.

(1) However, the methods section should be given some attention: Make different sections: 2.1.1 Questionnaires on PA and 2.1.2 SB in the self-report questionnaires (from line 131 onwards); 2.1.3 Forward-backward method, from line 139 onwards. Many statements onder the questionnaires should be briefly mentioned in introduction, but more in detail used for the discussion, but certainly not part of the methods section.

  • Thank you for this clarifying comment. This has been modified as suggested.

(2) The authors should also think about presenting some of the tabulated data in figure format, for the reader so get a better insight of what specifically is being compared. That would certainly also facilitate the writing of the results section.

  • Thank you. We considered this suggestion carefully but decided to keep the results in the table format. Each table includes several results, and we feel that presenting them with a figure, in an informative manner, would be challenging. One table would require several figures that would increase the total number of tables and figures too much.

(3) In the discussion, the authors should address more in detail the low number of participants in relation to drawing general conclusion. As well as compare the number of participants to other comparable studies, and discuss the gender fraction and how that contributed to attitudes and interpretations.

  • Thank you. We have added the low number of participants as one of the limitations of the study.

More specific edits/suggestions are presented below:

Introduction:

Lines 30-33 should read: ‘Low levels of physical activity (PA) and high sedentary behavior (SB) are recognized combined as widely devastating global public health problems that require drastic actions [1]. Physical inactivity i.e., not meeting the current PA guidelines [2], is harmful to an individual’s health and wellbeing, and increases the risk for many adverse health conditions…’

  • Thank you for correcting this. These sentences have been re-written.

Lines 37-39 should include a comma: ‘SB is a risk factor for all-cause and cardiovascular mortality, incident cardiovascular disease, and type 2 diabetes [4], partly independently of PA [5, 6].’

  • Thank you, we have added the comma.

Add citations to lines 57/59.

  • Thank you for asking clarification. We have added the references Sattler et al. (2016) and Prince et al. (2018) here.

Add citations to lines 61/64 and first part of 64/65 ‘…methods and algorhythms’.

Thank you for this comment. We have added the reference Sattler et al. (2016).

Add citations to lines 68/69 ‘…when monitoring PA and SB nationally’

  • Thank you for noticing this missing reference. We have added the reference Sember et al. (2020) here.

Add citation to lines 72/76.

  • Thank you. We have added the reference Sember et al. (2020) here as well.

Replace in line 79 ‘this’ for ‘the current’

  • Thank you for this correction. The word this has been replaced by the word current.

Replace in line 81/82 ‘their data with each other’ by ‘various PAQ data’.

  • Thank you for this notion. This has been replaced as suggested.

Write in line 83 ‘Additionally, the Eurobarometer questionnaire was included in the present study, since …’

  • Thank you for clarifying this. This has been revised as suggested.

Methods:

Line 87, replace: ‘The current investigation is part of the Finnish….’

  • Thank you! This has been replaced as suggested.

Lines 106/107 ‘but IPAQ-SF seems to overreport PA when compared to device-based measurement, which indicates poor criterion validity [16]’, perhaps belongs in the discussion.

  • Thank you for noticing this. This has been revised as suggested.

Lines 107/108 ‘VPA and walking showed the highest correlation with device-based measurements [16]’ should be part of the introduction.

  • Thank you for pointing this out. This has been moved to the introduction as suggested.

Line 116 ‘Additionally, one item assesses sitting during a typical day [20].’ Should be under section 2.1.2.

  • Thank you for detecting this. This has been moved under a new section 2.1.2.

Line 117 ‘GPAQ has shown moderate to very good test–retest reliability’ should be part of the discussion.

  • Thank you for identifying this. This has been moved to the discussion as suggested.

Lines 117/118 ‘but validity against device-based measurements has been poor or fair [21]’ should be part of the introduction.

  • Thank you. This has been moved to the introduction as suggested.

Line 124 ‘Test-retest 124 reliability of EHIS-PAQ has been moderate to high’ should be part of the discussion.

  • Thank you for spotting this. This has been moved to the discussion as suggested.

Lines 125/126 ‘but validity against device-based measurement is low [23]’ should be part of the introduction.

  • Thank you for noticing this. This has been moved to the introduction as suggested.

Make different sections: 2.1.1 Questionnaires on PA and 2.1.2 SB in the self-report questionnaires (from line 131 onwards); 2.1.3 Forward-backward method, from line 139 onwards. Many statements onder the questionnaires should be briefly mentioned in introduction, but more in detail used for the discussion, but certainly not part of the methods section.

  • Thank you for this clarifying comment. Different sections have been made as suggested.

Results

Line 242 instead of ‘than the’ use ‘as compared to the’

  • Thank you for noticing this language error. This has been revised accordingly.

Discussion

Line 261, remove ‘with each other’ to ‘PAQs showed good correlations for both sitting time and VPA’.

  • Thank you for this suggestion. This has been revised accordingly.

Round 2

Reviewer 1 Report

Comments and Suggestions for Authors

Thank you for your extensive revisions. There are only a few additional comments regarding the limitations of your study and clarification about the participant profile and base sizes in your results:

MATERIALS AND METHODS:

-        Please clarify the source(s) of your convenience sample and nature of incentive

  • Thank you for asking clarification for this. The sources of the sample have now been specified. No incentives were used, and this point has been added to the text (lines 97-99).

REVIEWER RESPONSE: There is a potential bias of your participants being being “neighbors, relatives, or friends of the UKK Institute employees”. Please include this in your limitations.

-        Please outline your power analysis to justify your sample size

  • Thank you for asking this important issue. Unfortunately, we did not perform any formal power analysis. The original purpose was to analyze data from several countries (about 70-80 participants per country) together which would have provided a larger sample size, but coexisting COVID-19 undermined the data collections and timetables in most countries, and therefore we decided to focus on the Finnish data with a sample size of 62 adults. Sample sizes of 40 or higher providing a broad variation in data are generally considered adequate to reveal meaningful differences between two different methods.

REVIEWER RESPONSE: Please validate this statement “Sample sizes of 40 or higher providing a broad variation in data are generally considered adequate to reveal meaningful differences between two different methods” with a reference.

RESULTS:

REVIEWER RESPONSE: A table of participant profile is needed outlining the sex, age, characteristics (e.g. bmi group), including numbers of different groups (e.g. friends, relatives, employees etc).

-        Lines 196-197: Please articulate the proportion underweight, normal, obese classes I to III per sex

  • These proportions have been added to the text (lines 222-225).

REVIEWER RESPONSE: Since your participant excludes the underweight obese groups (i.e. 1 underweight female is not representative) and is not a representative sample so please include this as a weakness of your study.

-        Tables 2 and 3: The 95% CI values are very broad for a number of your measures. Please explain these very large CI’s

  • Thank you for noticing this. Large CIs are based on the answers for different questions. Respondents’ interpretation about their PA and SB varies between the questionnaires and time points.

REVIEWER RESPONSE: Thank you for your response but these large CIs seem to be representative of your low sample sizes and should be discussed in as limitations of your study.

Please include the sample bases in your results tables as it was unclear if there was any missing data associated with the results presented.

DISCUSSION:

-        Please include your limitations such as you collected a convenience sample yet your statistical methods assumed a simple random sample was collected.

  • Thank you for asking clarification for this. Convenience sample has been mentioned as one of the limitations of the study (line 382). We have now calculated the Bootstrapped CIs which does not include any assumptions for distribution.

REVIEWER RESPONSE: Please see comments above regarding further weaknesses of your study.

Author Response

Reviewer 1

Thank you for your extensive revisions. There are only a few additional comments regarding the limitations of your study and clarification about the participant profile and base sizes in your results:

MATERIALS AND METHODS:

-        Please clarify the source(s) of your convenience sample and nature of incentive

  • Thank you for asking clarification for this. The sources of the sample have now been specified. No incentives were used, and this point has been added to the text (lines 97-99).

REVIEWER RESPONSE: There is a potential bias of your participants being being “neighbors, relatives, or friends of the UKK Institute employees”. Please include this in your limitations.

- Thank you for asking clarification for this. We have specified the convenience sample in lines 92-95 to indicate that the sample was collected nearby city of Tampere, Finland (e.g. among participants being involved in another study of the UKK Institute and/or being neighbors, relatives, or friends of the UKK Institute employees). Several employees of the UKK Institute participated to the recruitment of the participants with wide age range. Convenience sample has been mentioned as one of the limitations of the study, and we have added “non-representative and relatively small convenience sample” in line 368 to indicate the nature of the sample.

-        Please outline your power analysis to justify your sample size

  • Thank you for asking this important issue. Unfortunately, we did not perform any formal power analysis. The original purpose was to analyze data from several countries (about 70-80 participants per country) together which would have provided a larger sample size, but coexisting COVID-19 undermined the data collections and timetables in most countries, and therefore we decided to focus on the Finnish data with a sample size of 62 adults. Sample sizes of 40 or higher providing a broad variation in data are generally considered adequate to reveal meaningful differences between two different methods.

REVIEWER RESPONSE: Please validate this statement “Sample sizes of 40 or higher providing a broad variation in data are generally considered adequate to reveal meaningful differences between two different methods” with a reference.

  • The statement that sample sizes 40 or higher are adequate for method comparison studies is basically anecdotal but commonly adopted among researchers. However, upon your request and to be specific, we have now added the following sentence with a reference to the Discussion (lines 371-373) where we mention the sample size as a study limitation: “However, according to Altman [47] a sample size of at least 50 is considered adequate for a method comparison study”.

RESULTS:

REVIEWER RESPONSE: A table of participant profile is needed outlining the sex, age, characteristics (e.g. bmi group), including numbers of different groups (e.g. friends, relatives, employees etc).

- Thank you for asking this modification. We have added a new table 2 presenting this information and modified the results text (lines 200-207) accordingly. Unfortunately, we do not have numbers for these different groups. Convenience sample is always non-representative, and this has been mentioned as one of the limitations of the study (line 368).

-        Lines 196-197: Please articulate the proportion underweight, normal, obese classes I to III per sex

  • These proportions have been added to the text (lines 222-225).

REVIEWER RESPONSE: Since your participant excludes the underweight obese groups (i.e. 1 underweight female is not representative) and is not a representative sample so please include this as a weakness of your study.

- Thank you. Since the study is based on a convenience sample, it is non-representative. This has been mentioned in the limitations of the study (line 368).

-        Tables 2 and 3: The 95% CI values are very broad for a number of your measures. Please explain these very large CI’s

  • Thank you for noticing this. Large CIs are based on the answers for different questions. Respondents’ interpretation about their PA and SB varies between the questionnaires and time points.

REVIEWER RESPONSE: Thank you for your response but these large CIs seem to be representative of your low sample sizes and should be discussed in as limitations of your study.

Please include the sample bases in your results tables as it was unclear if there was any missing data associated with the results presented.

  • Thank you for pointing this out. Sample size affects CIs and small sample size means larger CIs. Small sample size has been mentioned as one of the limitations of the study: “The weaknesses of the study include a non-representative and relatively small convenience sample compared to other corresponding studies with approximately 100-300 participants [21, 35, 42, 45]. Larger sample size would likely have given more accurate results, for example in terms of narrower CIs.” (lines 367-371). We have added numbers of missing data to the table 3.

DISCUSSION:

-        Please include your limitations such as you collected a convenience sample yet your statistical methods assumed a simple random sample was collected.

  • Thank you for asking clarification for this. Convenience sample has been mentioned as one of the limitations of the study (line 382). We have now calculated the Bootstrapped CIs which does not include any assumptions for distribution.

REVIEWER RESPONSE: Please see comments above regarding further weaknesses of your study.

  • Thank you, the weaknesses of the study have been revised according to your comments (lines 367-375).

Reviewer 2 Report

Comments and Suggestions for Authors

The authors have changed the manuscript as suggested previously, adding subsections in the methods and moving various sentences to more appropriate positions, while including additional requested citations. Presenting the tabulated results as well in figure format is understandably a challenging task that has not been granted, but might have additionally clarified the comparisons of the manuscript. Nevertheless, the discussion of results and the manuscript's limitations has been handled appropriate.

Author Response

Reviewer 2

The authors have changed the manuscript as suggested previously, adding subsections in the methods and moving various sentences to more appropriate positions, while including additional requested citations. Presenting the tabulated results as well in figure format is understandably a challenging task that has not been granted, but might have additionally clarified the comparisons of the manuscript. Nevertheless, the discussion of results and the manuscript's limitations has been handled appropriate.

  • Thank you very much.